# The Impact of Selected Pretreatment Procedures on Iron Dissolution from Metallic Iron Specimens Used in Water Treatment

**Rui Hu** [1], **Arnaud Igor Ndé-Tchoupé** [2], **Mesia Lufingo** [3], **Minhui Xiao** [1], **Achille Nassi** [2], **Chicgoua Noubactep** [4,*] and **Karoli N. Njau** [3]

1   School of Earth Science and Engineering, Hohai University, Fo Cheng Xi Road 8, Nanjing 211100, China; rhu@hhu.edu.cn (R.H.); xiaominhui@hhu.edu.cn (M.X.)
2   Department of Chemistry, Faculty of Sciences, University of Douala, B.P. 24157 Douala, Cameroon; ndetchoupe@gmail.com (A.I.N.-T.); achillen@yahoo.fr (A.N.)
3   Department of Water and Environmental Science and Engineering, Nelson Mandela African Institution of Science and Technology, Arusha P.O. Box 447, Tanzania; lufingom@nm-aist.ac.tz (M.L.); karoli.njau@nm-aist.ac.tz (K.N.N.)
4   Department of Applied Geology, Universität Göttingen, Goldschmidtstraße 3, D-37077 Göttingen, Germany
*   Correspondence: cnoubac@gwdg.de

**Abstract:** Studies were undertaken to determine the reasons why published information regarding the efficiency of metallic iron (Fe$^0$) for water treatment is conflicting and even confusing. The reactivity of eight Fe$^0$ materials was characterized by Fe dissolution in a dilute solution of ethylenediaminetetraacetate (Na$_2$–EDTA; 2 mM). Both batch (4 days) and column (100 days) experiments were used. A total of 30 different systems were characterized for the extent of Fe release in EDTA. The effects of Fe$^0$ type (granular iron, iron nails and steel wool) and pretreatment procedure (socking in acetone, EDTA, H$_2$O, HCl and NaCl for 17 h) were assessed. The results roughly show an increased iron dissolution with increasing reactive sites (decreasing particle size: wool > filings > nails), but there were large differences between materials from the same group. The main output of this work is that available results are hardly comparable as they were achieved under very different experimental conditions. A conceptual framework is presented for future research directed towards a more processed understanding.

**Keywords:** contaminant removal; electrochemical reaction; operational parameters; zerovalent iron

## 1. Introduction

Metallic iron (Fe$^0$) has been used in the water treatment industry for more than 140 years [1–20]. Fe$^0$ filters are reported to be common place in Europe in the 1860s [19]. A real breakthrough for household Fe$^0$ water filters was realized with the Bischof Process using spongy iron (sponge iron) as the reactive material [2,3,8,18]. The first large scale application of Fe$^0$ for safe water provision is likely the one tested and installed in the city of Antwerp (Belgium) in the early 1880s [2,8,18–20]. Here, a filter made of sponge iron and sand (volumetric Fe$^0$:sand = 1:3) could successfully work for 18 months with little maintenance before experiencing clogging. Difficulties to manage clogging resulted in the development of the Anderson Process in which polluted water is equilibrated with iron filings within a "revolving purifier" [2,8,20,21]. The Anderson Process is, in essence, a coagulation event in which coagulants (iron hydroxides) are generated in situ [22–24].

During the past 30 years, Fe$^0$ (largely termed zero-valent iron (ZVI)) has been rediscovered and tested for the removal of several classes of contaminants including azo dyes [25,26], chlorinated

organic [27,28], fluoride [29–31], heavy metals [32,33], nitrate [34,35], pathogens [36–38] and radionuclides [32,39]. The desalination of water using $Fe^0$ has also been reported [40–43]. The two key advantages of $Fe^0$ are its affordability and its universal availability (iron nails, scrap iron and steel wool) [9,10,44,45]. It is commonly reported that the major drawback of $Fe^0$-based technologies for water treatment is the low intrinsic reactivity of granular materials [28,33,46]. This situation is said to be aggravated by the inherent generation of an oxide scale at the $Fe^0$ surface (passivation). Countermeasures to overcome $Fe^0$ passivation were recently reviewed by Guan et al. [28] and further discussed by Noubactep [47]. It is recalled that, considering passivation as a "curse" contradicts the evidence that $Fe^0$-based subsurface permeable barriers have been successfully working for more than one decade [48–50]. On the other hand, increased "passivation" should have occurred in the spongy iron filters in Antwerpen as well [2,19]. Clarifying this contradiction is certainly a progress for the whole $Fe^0$ technology. Clearly, an understanding of the role of "passivation" in the process of decontamination using $Fe^0$ should be useful for enhancing the system's efficiency in practice [30,31,47].

There is an agreement on the evidence that using $Fe^0$ for water treatment and environmental remediation is "putting corrosion to use" [51–53]. There is otherwise a contradiction on practically any other aspect concerning the $Fe^0/H_2O$ system, including the reaction mechanisms and factors determining the long-term efficiency of such systems [9,28,47,54–58]. A key reason for this is the large variability of experimental conditions used in testing $Fe^0$ materials [58–62]. The main influencing operational parameter seems to be $Fe^0$ itself [58,61,63,64]. In fact, data from the open corrosion literature demonstrate that at pH > 4.5, the extent of iron corrosion depends on the relative kinetics of iron oxidative dissolution (corrosion) and of the precipitation of resulting iron hydroxides and oxides (oxide-scale formation) [65–69]. Accordingly, when iron hydroxide precipitates at the $Fe^0$ surface or in its vicinity, it slows down the further dissolution process by partly or entirely shielding the $Fe^0$ surface for the species involved in the corrosion process, including dissolved oxygen and reducible contaminants [70–72]. The oxide scale is, in essence, a diffusion barrier, and its shielding property depends on its porosity and its adherence to the $Fe^0$ surface [66–68]. The oxide scale interacts with contaminants as well and complicates the understanding of the $Fe^0/H_2O$ system [73–75].

Many interrelated factors affect the formation and transformation of an oxide scale (e.g., $Fe^0$ size, surface roughness and water chemistry). The most important influencing factor is the water chemistry, including the presence of co-solutes (and ligand), the pH value, the temperature and the water flow velocity [66–68]. Water treatment using $Fe^0$ mostly occurs at pH > 5.0, which corresponds to the range of quantitative precipitation of iron oxides [27]. In this pH range, it is very difficult to correlate the corrosion and the decontamination processes because of the omnipresence of the oxide scale [73–75]. The oxide scale is generated and transformed in a dynamic process ("rust never rests") [66,76]. Around 2005, two research groups independently presented $Fe^0$ dissolution in a dilute solution of ethylenediaminetetraacetic acid (EDTA) as a powerful tool to characterize the initial kinetics of iron corrosion (EDTA method) [77,78]. Herein, the absence of contaminants and iron corrosion products (FeCPs) facilitates the discussion of the investigated parameters [60,61,79,80]. The EDTA method was already used to demonstrate the impact of selected operational conditions on the investigations of processes occurring in $Fe^0/H_2O$ systems. In particular, it was demonstrated that quiescent batch experiments are the best conditions to investigate diffusion-controlled processes occurring within the dynamic oxide scale [33,60,61].

The present work aims at using the EDTA method to characterize the impact of some relevant operational conditions on the efficiency of $Fe^0$ materials and to discuss their possible impacts in real systems. It is expected to bring clarity on some controversies and thus, shape appropriate investigations for the design of reliable and more sustainable $Fe^0$-based systems. The investigated parameters include material pretreatment and $Fe^0$ type (iron filing, iron nails and steel wool). The characterized pretreatments included soaking 4 different $Fe^0$ specimens in 5 different solutions prior to use in batch dissolution experiments. The iron dissolution from $Fe^0$ was further characterized in column leaching experiments using iron nails. The results are comparatively discussed.

## 2. Materials and Methods

### 2.1. Solutions

Based on previous works [60,61,77], a working EDTA solution of 0.002 M (or 2mM) was used in this study. The working-solution was prepared from a commercial disodium salt ($Na_2$–EDTA). A standard iron solution (1000 mg $L^{-1}$) from Baker JT® was used to calibrate the Spectrophotometer. All other chemicals used were of analytical grade. In preparation for the spectrophotometric analysis, ascorbic acid was used to reduce $Fe^{III}$–EDTA in solution to $Fe^{II}$–EDTA. The reagent used for $Fe^{II}$ complexation prior to spectrophotometric determination was 1,10-orthophenanthroline (ACROS Organics). Other chemicals used in this study included acetone, L(+)-ascorbic acid, L-ascorbic acid sodium salt and sodium chloride.

### 2.2. $Fe^0$ materials and sand

A total of 8 $Fe^0$ materials were obtained from various sources (Figure 1) in different forms and grain sizes. The main characteristics of these materials including form, grain size and elemental composition are summarised in Table 1. Roughly, the tested materials are four iron nails (FeN1, FeN2, FeN3 and FeN4), one steel wool (SW) and 3 granular $Fe^0$ (GI1, GI2 and GI3). It was the explicit objective of this study to compare the reactivity of the tested materials in their typical state (and form) in which they might be used for field applications. All materials were, thus, used in the "as received" state.

The used sand was a commercial material for aviculture ("Aquarienkies" sand from Quarzverpackungswerk Rosnerski Königslutter/Germany). Aquarienkies was used as received without any further pretreatment. The particle size was between 2.0 and 4.0 mm. Sand is used as a filling material in the column studies.

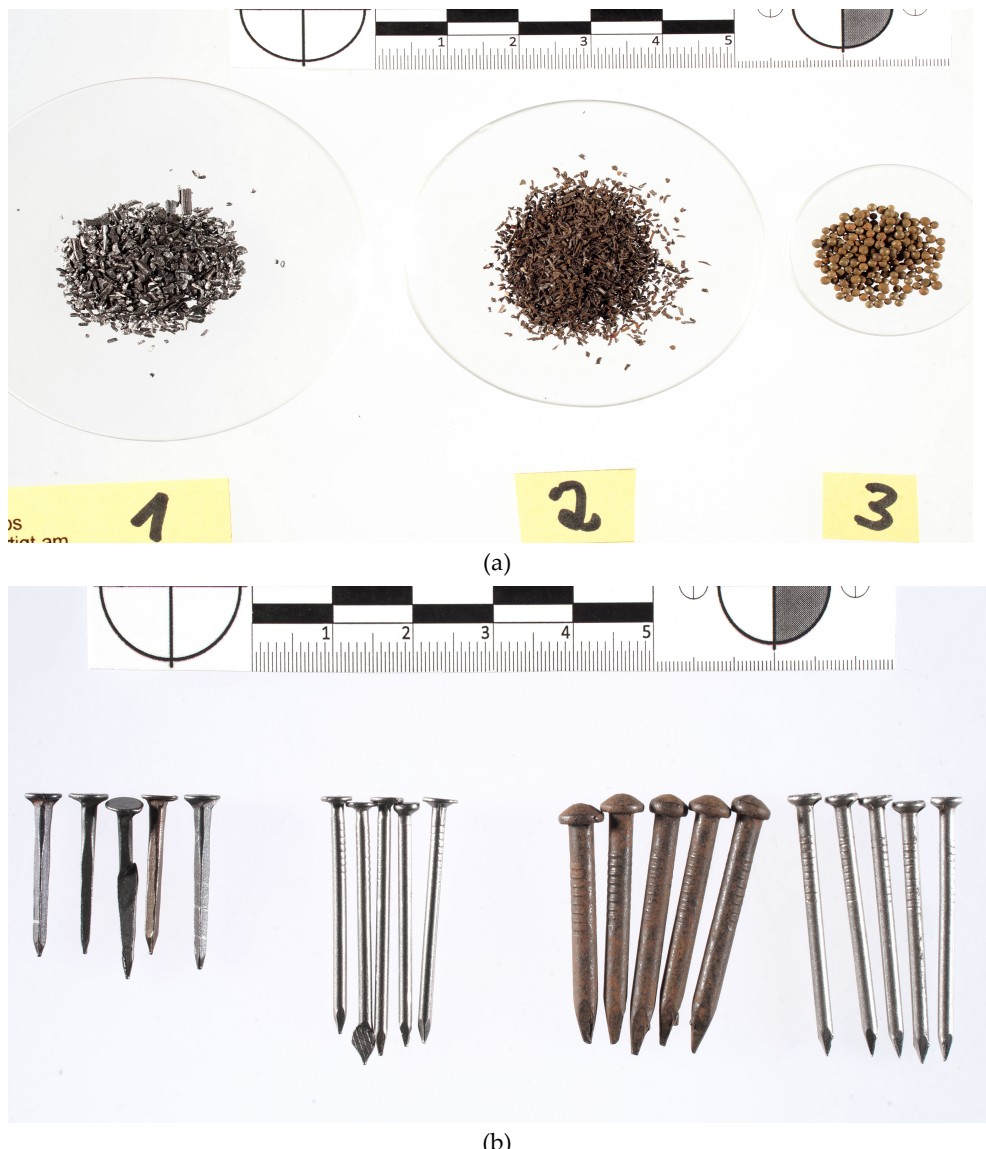

**Figure 1.** Photographs of the used Fe$^0$ specimens: (**a**) the three granular material (GI1, GI2 and GI3 from left to right; 2.0 g of each) and (**b**) the four iron nails (FeN1, FeN2, FeN3 and FeN4 from left to right; each 5 nails). The steel wool specimen was not considered.

**Table 1.** The origin, name and main characteristics of the tested Fe$^0$ materials. For the specimens tested in the column experiments, "m" (g) is the used mass, N the corresponding number of particles and P (%) is the percent Fe leaching at the end of the leaching column experiments. "n.a." stands for not available. The N value was determined for spherical GI3 because the filings are difficult to count. Accordingly, m values for GI3 are just given to evaluate the N value for GI1.

| Origin | Original Denotation | Trade Mark | Type | Code | m (g) | N (-) | P (%) |
|---|---|---|---|---|---|---|---|
| Germany | IpuTech GmbH | FG 1000/3000 | filings | GI1 | 2.19 | >232 | 22.8 |
| Germany | MAZ mbH | Sorte 69 | scrap iron | GI2 | 2.00 | >232 | n.a. |
| Germany | Hartgusstrahlmittel | Würth | filings | GI3 | 2.00 | 232 | n.a. |
| Kenya | Shivkrupa Investements | Champion | steel wool | SW | 2.22 | n.a. | 37.5 |
| China | Qingdao | Three stars | nails | FeN1 | 2.21 | 6 | 6.1 |
| Unknown | n.s | Santoplus | nails | FeN2 | 2.28 | 4 | 9.3 |
| Unknown | n.s | Caserio | nails | FeN3 | 2.06 | 2 | 4.0 |
| Cameroon | Prometal | Tik | nails | FeN4 | 2.32 | 3 | 5.3 |

## 2.3. Experimental Procedure

### 2.3.1. $Fe^0$ Pretreatment

Six different pretreatment methods were tested in this study; an accompanying experiment with non pretreated $Fe^0$ was performed. The tested pretreatment consisted in socking the weighed $Fe^0$ in the following solutions for 17 hours: (i) tap water ($H_2O$), (ii) NaCl (1 M), (iii) HCl (0.1 M), (iv) commercial acetone and (v) EDTA (2 mM). The pretreated materials were further washed three times with tap water prior to the addition of the EDTA solution. The EDTA solution was added to the wet pretreated materials. Four $Fe°$ materials were involved in this study: iron filings with metallic glaze (iPuTech; GI1), a rusted scrap iron with a rough surface (S69; GI2), a rusted spherical shape (GI3) and a steel wool material (SW) (Figure 1).

### 2.3.2. Iron Dissolution Studies

Iron dissolution was used to characterize (i) the reactivity of the tested $Fe^0$ [60,80] and (ii) the impact of different pretreatments on the (initial) efficiency of the $Fe^0$ materials [60].

Batch Experiments

The iron dissolution was initiated by the addition of 0.1 g of each GI or 0.01 g of SW to 50 mL of a 2.0 mmol $L^{-1}$ EDTA solution. Each reaction was run for 96 h (4 days) using narrow 70 mL glass beakers to hold the solutions [60,61,80]. The reacting samples were left undisturbed on the laboratory bench for the duration of the experimental period and were shielded from direct sunlight to minimize $Fe^{III}$–EDTA photodegradation [61].

Column Experiments

The laboratory scale glass columns were operated in downflow mode. Six glass columns (40 cm long, 3.0 cm inner diameter) were used. The columns were packed to one half with sand. Each column contains about 2.0 g of a different $Fe^0$ material (the four FeNs, GI2 and SW) in its most upper part. The packed columns were not further characterized as only the kinetics, and the extent of iron oxidative dissolution in EDTA were the targets [79]. The influent EDTA solution (2 mM) was intermittently added to the free half of the column five times per week (Monday to Friday). Thus, the solutions were allowed to equilibrate in the columns for about 24 hours during the week and 72 hours during the weekend. The experiment was performed at room temperature (22 $\pm$ 3 °C) from June 19 to September 17, 2018 (100 days). During these 100 days, 64 leaching events were realized. Each leaching event corresponds to some 390 mL of the EDTA solution. The leachates were analysed for iron, and their exact volumes were recorded.

## 2.4. Analytical Method

The aqueous iron concentration was determined with a Varian Cary 50 UV-vis spectrophotometer (Agilent Technologies, Santa Clara, CA, USA), using a wavelength of 510 nm and following the 1,10-orthophenanthroline method [81]. The instrument was calibrated for iron concentration $\leq$ 10 mg $L^{-1}$.

## 3. Results and Discussion

All results are expressed in terms of iron concentration (mg $L^{-1}$). For the column experiments, the mass of leached iron is additionally given. Assuming a 1:1 Fe–EDTA complexation, working with an initial EDTA concentration of 2 mM implies a solution saturation at [Fe] = 112 mg $L^{-1}$. Overviewing the results, it is clear that this value has never been achieved. In the column experiments, the largest [Fe] value was obtained with SW. This means that no equilibrium stage and no saturation was achieved in this work [82]. This fact rationalizes why no attempt was done to model the results. Even the $k_{EDTA}$

values [61,77] were not derived. This evidence makes the results purely qualitative and corresponds to the objectives of this study.

## 3.1. Reactivity of $Fe^0$ Materials

The reactivity of "as received materials" were characterized in the batch and column studies (Figures 2 and 3)

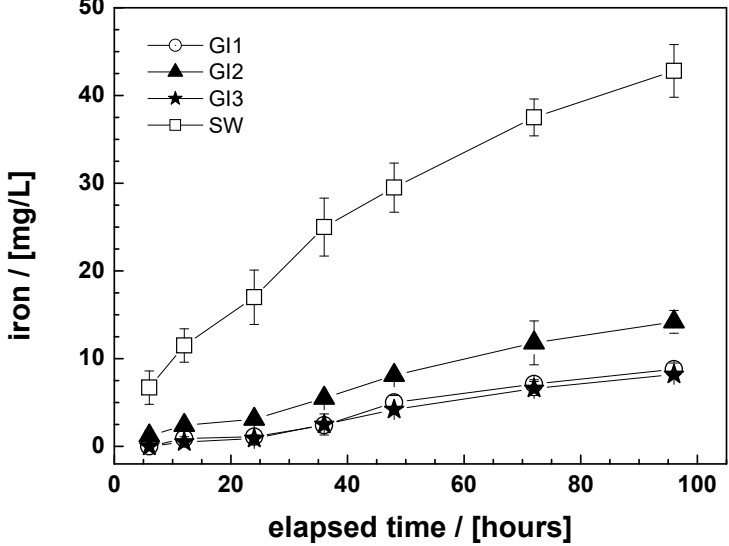

**Figure 2.** The iron release (mg $L^{-1}$) from the four tested $Fe^0$ materials after four days (96 hours) in the batch experiments. The experimental conditions were [GI] = 2 g $L^{-1}$, [SW] = 0.2 g $L^{-1}$, V = 50 mL and [EDTA] = 2 mM. The lines are not fitting functions; they simply connect the points to facilitate a visualization.

### 3.1.1. Batch Experiments

Figure 2 summarized the results of the dissolution of the four $Fe^0$ materials used in batch studies. It is evident that SW was the most reactive material. The derived order of reactivity is SW > GI2 > GI1 = GI3. It should be kept in mind that only 0.01 g of SW was used (against 0.1 g GIs). This choice is based on recent results [80] demonstrating that 0.01 g of SW enables the achievement of a linear regression in the time frame where 0.1 g of GI is needed. A surprising aspect is the similitude of the reactivity of GI1 and GI3. Both materials were selected herein based on their different reactivity as concordantly documented by Noubactep et al. [77] and Btatkeu-K et al. [61]. The additional aspect to be considered is that the same specimen of spherical GI3 is used in the three works while GI1 used herein was recently purchased from iPuTech (GmbH). Thus, GI1 is not strictly the one used in the other two works. On the other hand, GI3 corrosion under laboratory conditions has continued for five more years, producing fines that have been shown to quickly dissolve in EDTA as well [60,77]. The results of the pretreatment experiments will help to clarify this issue (Section 3.2).

The main feature from Figure 2 is that $Fe^0$ materials are of various reactivities. Thus, material selection for an application should be site-specific. It cannot be simply stated that SW is better or worse than GI like it is currently done in the $Fe^0$ remediation literature [83–85]. For example, Allred [84] recently evaluated the suitability of 58 industrial aggregates (products and by-products), including 8 $Fe^0$ specimens as filter materials for the treatment of agricultural drainage water. $NO_3^-$, $PO_4^{3-}$ and atrazine were used as model pollutants. The experimental systems were thoroughly mixed at 20 rpm for 24 h. The authors excellently acknowledged that more extensive testing experiments are needed for a complete evaluation of the suitability of the tested materials for drainage water treatment. However, it is evident that 24 h is too short for testing such reactive materials that, in situ, generate

contaminant scavengers. Moreover, the kinetics of the scavenger's generation (i) has not yet been properly characterized and (ii) is not linear [86,87]. Accordingly, it is a thinking mistake to compare the efficiency of pure adsorbents and of $Fe^0$ materials under the same operational conditions. By doing so, the specificity of $Fe^0$, making it applicable for decades, is not addressed [46–50].

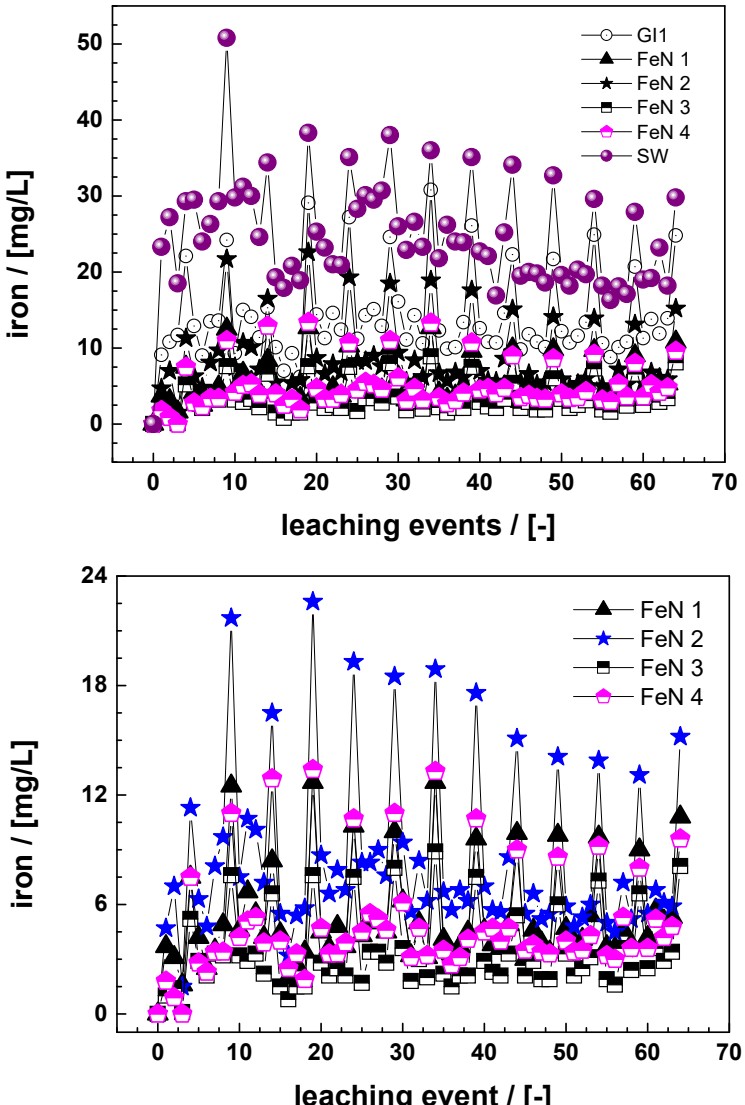

**Figure 3.** The iron release (mg $L^{-1}$) in the column experiments after 64 leaching events: (**a**) all six tested $Fe^0$ materials and (**b**) the four iron nails. The experimental conditions were $m_{iron}$ = 2 g, $V_{EDTA}$ = 390 mL and [EDTA] = 2 mM. The lines are not fitting functions; they simply connect the points to facilitate a visualization.

The approach used by Allred [84] is commonplace in the recent literature and was introduced already in the 1990s [6,88,89], resulting in disqualifying $Fe^0$ for water treatment [89] or recommending it [6,88]. Since then, a legitimate scepticism on the efficiency of $Fe^0$ for water treatment is available within the remediation community [90,91]. This trend still exists despite some success stories both for safe drinking water supply [92–94] and groundwater remediation [48–50,95]. Eradicating this scepticism will be a gain for the whole scientific community. In particular, the established suitability of commercial iron nails [94] for decentralized safe drinking water supply is presented as a proof of the potential of $Fe^0$-based systems to assure universal access to safe drinking water [10,96,97].

The $Fe^0$ materials tested by Allred [84] were from three different origins and included a carbon reduced iron, a hydrogen reduced iron, a porous iron composite, two scrap iron specimens and three sulfur modified iron (SMI) specimens. The results revealed (i) that all eight materials were very efficient at removing $PO_4^{3-}$ (>94%) and (ii) that only the SMI specimens achieved more than 50% $NO_3^-$ removal and the two reduced irons exhibited the worst efficiency for atrazine. These results clearly showed that it is impossible to recommend any material for water treatment based on the results of screening tests. For the materials tested herein, it can be postulated that there are situations in which SW is suitable and others where GI1, GI2 or GI3 are better suitable. The long history of using $Fe^0$ for water treatment suggests that suitable $Fe^0$ materials were mostly found for reported success stories, but the rational for their selection has not been really realized [58,63,64]. Even today, it is current to consider GI or SW as a class of material compared to nano-$Fe^0$, iron nails or bimetallics. As demonstrated herein (Figure 2), GI is not a homogeneous class of materials; Hildebrant [80] demonstrated that SWs are also not homogeneous in reactivity. While this situation is known for a long time [59,98], little is done to introduce a reliable test to characterize the intrinsic reactivity of $Fe^0$ materials [58,61,63,64]. Moreover, some few tests have been suggested and have not been really independently used [58,63,64,99]. The sole test that was used by at least one different research group is the $H_2$ evolution method [99]. However, $H_2$ evolution was not used in a developmental perspective, but the researchers have just slightly modified the original protocol [95,100,101]. For example, Velimirovic et al. [101] used $H_2$ measurements for up to 105 days not to characterize the intrinsic reactivity of 17 $Fe^0$ materials but to derive their corrosion rate and to characterize the effect of aquifer materials thereon. Velimirovic et al. [101] positively correlated the $Fe^0$ corrosion rate, particle sizes (specific surface area) and degradation rates of chlorinated aliphatic hydrocarbons. Generally, their results demonstrated that $Fe^0$ specimens with faster corrosion rates ($H_2$ evolution) exhibit higher degradation rates. This intuitive general trend was not changed in the presence of aquifer material or while using real groundwater. In other words, despite longer experimental durations compared to Allred [84] (1 vs. 105 days) and the derivation of corrosion rates, Velimirovic et al. [101] have not presented any transferable results.

The apparent corrosion rate (e.g., mmol $kg^{-1}$ $d^{-1}$) presented by Reference [101] and the derived lifetime of $Fe^0$ particles (11 to 247 years) have not considered their particle size. Results corresponding to the expected service life of $Fe^0$ walls up to three decades and could have mistakenly validated the used model. However, no accurate model can be presented without considering the complete analysis of the system [102,103]. Considering the size of $Fe^0$ materials in estimating the service life of remediation systems is mandatory. This is even essential because the assumed uniform corrosion of individual particles is already an oversimplification [103]. Accordingly, the apparent corrosion rate should be related to individual particles (e.g., mmol $kg^{-1}$ $d^{-1}$ $particle^{-1}$), and the time-dependent law of variation with the changes in the particle size should be considered [86].

The presentation until now has clearly demonstrated that the current $Fe^0$ literature is full of qualitative results. Moreover, it is not possible to trace the origin of the controversy from independent reports. The present contribution does not aim at presenting any new results but rather at contributing to convince the community of the urgency of recognizing that better investigations are needed. Generally, the authors consider that only a broad awareness on the origin of the confusion installed in the $Fe^0$ research community would initiate required changes. These changes have the potential to make $Fe^0$-based systems a universal solution for safe drinking water. The next section presents the results from the column experiments, specially designed to compare the reactivity of iron nails.

### 3.1.2. Column Experiments

Figure 3 shows that GI1 and SW exhibited markedly increased dissolution kinetics during the whole experiments than the four FeNs; see also the P values in Table 1. It is seen that the highest picks correspond to the weekend values (72 h of equilibration). These values were however always lower than the saturation value (112 mg $L^{-1}$). The N and P values in Table 1 tend to correlate with each

other: "the larger the number of particles making up the 2.0 g of $Fe^0$, the higher the P value". This rule of thumb is the idea behind using nano-$Fe^0$ in water treatment [104]. However, the P values also clearly show that smaller materials will be exhausted sooner. Clearly, while stressing the six materials under identical conditions, only 4.0 to 9.3% iron dissolves from the nails while 22.0% dissolves from the filings (GI1) and 38.0% from steel wool (SW). The corresponding order of reactivity is FeN3 < FeN4 < FeN1 < FeN2 < GI1 < SW. Section 3.1.1 has already demonstrated the GIs are of various reactivities. Hildebrant [80] recently established variabilities in the reactivity of SWs as well. The present results clearly show that iron nails are equally variable in reactivity. This observation is essential for the design of decentralized FeN-based water treatments as currently operated in arsenic-affected regions [94,105,106]. The difference in reactivity documented herein suggests that taking the intrinsic reactivity of FeN in designing water filters would yield more sustainable systems. This qualitative result can be regarded as a cornerstone for future investigations implying water pollutants, including methylene blue [107]. Is there any correlation between this initial dissolution in EDTA and the $Fe^0$ efficiency for decontamination? Only a large number of experiments performed under controlled conditions will enable a science-based answer to this question.

The next sections report on the impact of the $Fe^0$ pretreatment on the dissolution behaviour of Fe from the four materials in the batch systems.

### 3.2. Effect of Material Pretreatment

Table 2 summarizes the results for the five different pretreatment procedures for the four tested materials (24 systems). The pretreatment consisted of socking the weighed $Fe^0$ mass in a 50 mL treatment solution for 17 h (overnight). The results presented in Table 2 (P values) showed that no pretreatment procedure could enhance the reactivity of GI1, GI3 and SW. For GI2, HCl washing yielded 14% more iron dissolution than in the reference system (no treatment). On the other hand, taking the GI1 as reference system, the "Factor" values clearly demonstrated that GI2 and SW are more reactive than GI1; this observation corroborates the discussion in Section 3.1.1. GI1 was taken as a reference system because of its worldwide use [61,108]. Accordingly, situations are compared, where different materials and different pretreatment procedures are used. Clearly, 24 possible starting experimental conditions, collectively related to "$Fe^0$ for water treatment", are compared. Disregarding any mechanistic discussions, the results in Table 2 rationalize the current "dialogue of the deaf" within the $Fe^0$ research community [109]. The way out of this confusing situation goes exclusively through systematic investigations [110,111].

The "Factor" values show that, under tested conditions, a relative amount of 0.6 to 4.8 Fe is released into the individual systems. This is certainly the main feature of this work because there is currently no unified procedure to characterise the intrinsic reactivity of $Fe^0$-based materials [110]. Treatability studies are also performed under ill-defined conditions; and the results are compared using models that have been proven wrong a decade ago [55,112,113]. For example, the removal capacity of $Fe^0$ materials is usually presented in mg of contaminant per g of material in a context where nothing is known about the corrosion rate or the available amount of in situ generated adsorbents (scavengers). Using quiescent batch experiments herein, such a difference is documented in the 24 systems. Whether this difference is large or small is not discussed herein. The point is that 5.3 to 42.5 mg/L (0.3 to 2.1 mg in 50 mL) of Fe is dissolved from materials varying in forms, shapes and surface states. These parameters and others have been convincingly demonstrated significant in impacting the efficiency of $Fe^0$ remediation systems [58–60]. The question is how to compare the independent results.

**Table 2.** The corresponding iron concentrations after 96 h of equilibration with EDTA after various treatment procedures. "[Fe]" represents the standard deviation of individual triplicates. As a rule, the more reactive a material is under given conditions the bigger the iron concentration. The general conditions were initial pH = 5.2, [EDTA] = 2 mM, room temperature $23 \pm 2$ °C and $Fe^0$ mass loading $2$ g L$^{-1}$ for GI and 0.2 g L$^{-1}$ for SW. "Order" is the relative order of reactivity of individual materials as impacted by pretreatment. "Factor" is the relative reactivity of individual 24 systems relative to non-treated GI1. "P (%)" is the relative extent of Fe dissolution, taking the Fe concentration in the non-pretreated system as an operational reference.

| Material | Treatment | [Fe] (mg L$^{-1}$) | [Fe] (mg L$^{-1}$) | P (%) | Order (-) | Factor (-) |
|---|---|---|---|---|---|---|
| GI1 | None | 8.8 | 0.7 | 100 | 1 | 1.0 |
| | H$_2$O | 8.1 | 0.6 | 91 | 3 | 0.9 |
| | NaCl | 7.2 | 0.4 | 81 | 4 | 0.8 |
| | HCl | 8.5 | 1.4 | 96 | 2 | 1.0 |
| | Acetone | 5.8 | 0.3 | 66 | 6 | 0.7 |
| | EDTA | 6.1 | 0.4 | 69 | 5 | 0.7 |
| GI2 | None | 14.2 | 1.3 | 100 | 2 | 1.6 |
| | H$_2$O | 12.1 | 1.1 | 86 | 3 | 1.4 |
| | NaCl | 9.6 | 1.1 | 68 | 4 | 1.1 |
| | HCl | 16.1 | 1.5 | 114 | 1 | 1.8 |
| | Acetone | 9.2 | 1.0 | 65 | 5 | 1.0 |
| | EDTA | 8.1 | 0.7 | 57 | 6 | 0.9 |
| GI3 | None | 8.2 | 0.2 | 100 | 1 | 0.9 |
| | H$_2$O | 7.1 | 0.1 | 87 | 2 | 0.8 |
| | NaCl | 6.0 | 0.5 | 73 | 4 | 0.7 |
| | HCl | 6.4 | 1.0 | 78 | 3 | 0.7 |
| | Acetone | 5.4 | 0.4 | 65 | 6 | 0.6 |
| | EDTA | 5.4 | 0.4 | 66 | 5 | 0.6 |
| SW | None | 42.8 | 3.0 | 100 | 1 | 4.8 |
| | H$_2$O | 31.5 | 0.8 | 73 | 4 | 3.6 |
| | NaCl | 33.3 | 5.1 | 78 | 2 | 3.8 |
| | HCl | 25.8 | 0.3 | 60 | 6 | 2.9 |
| | Acetone | 30.3 | 1.3 | 71 | 5 | 3.4 |
| | EDTA | 32.2 | 4.6 | 75 | 3 | 3.6 |

The last important feature from Table 2 is that the order or reactivity observed in the reference system (GI3 < GI1 < GI2 < SW) is the same for the five types of pretreatments tested (Figure 4). Only the magnitude of iron dissolution changed. The extent of the Fe dissolution for the systems with H$_2$O, NaCl, acetone and EDTA are very close to each other and is markedly different from that of the EDTA pretreated one. For this reason, the H$_2$O and HCl systems are presented. The results in Figure 4 corroborate previous reports [33,60,74,75] that pretreatment procedures mainly modify the availability of iron corrosion products (FeCPs). While under moderate conditions, atmospheric FeCPs are washed away, under more severe condition (e.g., HCl washed) the external layers for the metal are dissolved. This situation implies that subsequent dissolution in EDTA addresses deeper $Fe^0$ layers that are difficult to "mine". This evidence explains why iron release from SW is lower after HCl washing and also why in the reference system (Figure 2), there is no reactivity difference between GI1 and GI3. The best differentiation of the reactivity of GI1 and GI3 corroborates this view that EDTA dissolved both atmospheric corrosion products and metallic iron. However, the relevance of pretreating $Fe^0$ before testing should be brought into question because the $Fe^0$ materials used in field situations are not commonly pretreated prior to emplacement. Even if materials were pretreated before implementation, surface oxides would rapidly form long before any significant quantity of contaminant inflow [60].

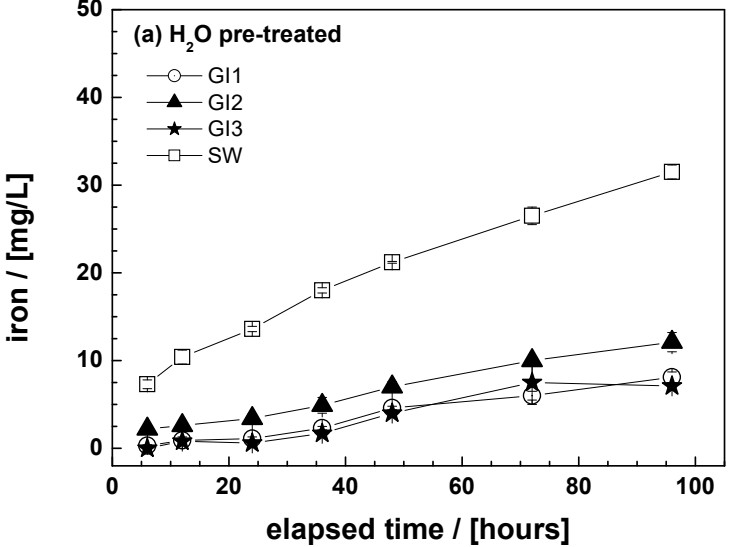

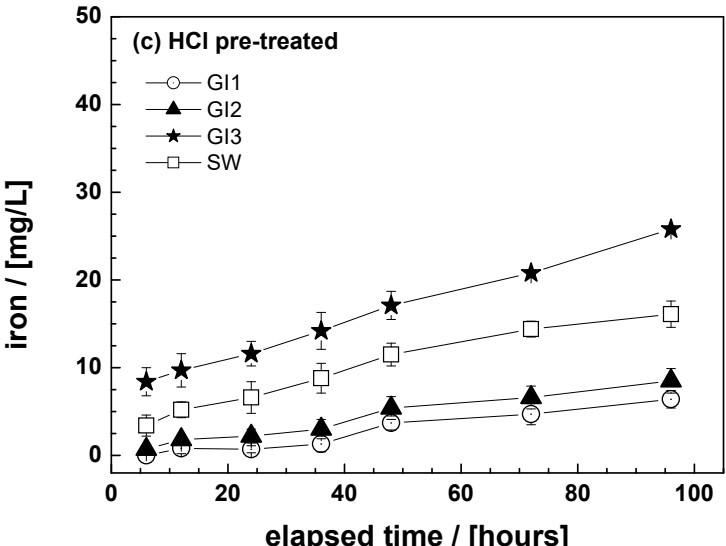

**Figure 4.** The iron release (mg L$^{-1}$) from the four tested Fe$^0$ materials after four days (96 hours) in the batch experiments as impacted by (**a**) H$_2$O pretreatment and (**b**) HCl pretreatment. The experimental conditions were [GI] = 2 g L$^{-1}$, [SW] = 0.2 g L$^{-1}$, V = 50 mL and [EDTA] = 2 mM. The lines are not fitting functions; they simply connect the points to facilitate a visualization.

*3.3. Discussion*

Metallic iron (Fe$^0$) is a potential reducing agent for many reducible species, including water (H$_2$O) and O$_2$ [15,26,56,57,114]. The redox potential of water is 0.00 V, making water a powerful oxidizing agent for Fe$^0$ (E$^0$ = −0.44 V). Indeed, aqueous iron corrosion (Equation (1)) is a stand-alone branch of scientific research with its own sections at several well-established peer-reviewed journals (e.g., Corrosion Science from Elsevier).

$$Fe^0 + 2\,H^+ \Rightarrow Fe^{2+} + H_2, \tag{1}$$

Equation (1) recalls that aqueous iron corrosion generates Fe$^{II}$ and H species which are stand-alone reducing agents [95]. For example, research on atmospheric corrosion over some three or four decades has established that Fe$^0$ is oxidized by water and Fe$^{II}$ by molecular O$_2$ [115,116]. This occurs because at

pH > 4.5, the $Fe^0$ surface is constantly shielded by a universal oxide scale, acting as a diffusion barrier for any dissolved species [67–72]. Because the oxide scale is constantly present at the $Fe^0$ surface, the discussion of the mechanism of contaminant removal has been challenging [33,55–57,72,111]. The results achieved herein contribute to clarify this dilemma because EDTA has no redox properties. Despite the accessibility of the $Fe^0$ surface to $O_2$ (no oxide scale), $O_2$ has to be diffusion-transported to the $Fe^0$ surface at the bottom of the reaction vessel. The diffusion is induced by the gradient of dissolved iron concentration while $H_2O$ is constantly present at the $Fe^0$ surface. Thus, iron is corroded by water, and the corrosion process is sustained by the consumption of $Fe^{II}$ for the $O_2$ reduction. Clearly, iron dissolution is an electrochemical process (electrons from $Fe^0$; Equation (1)), and $O_2$ reduction is a chemical reaction (electrons from $Fe^{II}$). The relation between $O_2$ reduction and $Fe^0$ oxidation is an indirect one going through the LeChatelier's Principle: accelerated corrosion by virtue of $Fe^{II}$ consumption.

Iron is corroded by water after Equation (1), and generated $Fe^{II}$ is complexed by EDTA and further oxidized to more stable $Fe^{III}$–EDTA complexes. By complexing $Fe^{II}$ and $Fe^{III}$ species, EDTA enables the observation of the further oxidation that would have not been possible because of the very low solubility of Fe species at pH > 4.5 implying particle cementation [35,78]. In column experiments, this observation could last for 100 days without any disturbance. It is wished that future investigations last for an even longer time, depending on the specific objectives. Now that the generation of the oxide scale and particle cementation are avoided to investigate the dissolution of $Fe^0$ for up to 100 days, the question arises how to use the results to understand long-term contaminant removal in real systems?

The past three decades have been a period of an uncoordinated search of inexpensive, reactive $Fe^0$ materials (including nano-$Fe^0$) [117,118]. The inherent generation of FeCPs has been regarded as a curse for the system. However, this trend was not univocal as steel wool has been independently developed for $PO_4$ remediation [85,119]. Whether $Fe^0$ is used for contaminant degradation, reduction or removal, its success depends on sustained corrosion. The key for sustained corrosion is known: no impermeable oxide scale at the $Fe^0$ surface [66–68]. The permeability of the oxide scale depends on the relative kinetics of (i) the iron dissolution and (ii) the formation of the oxide scale [67,68]. The named kinetics are also influenced by operational conditions ($O_2$ level, pH value, salinity and surface state of iron). The $Fe^0$ reactivity as characterized herein should be coupled to the efficiency of individuals for contaminant removal under various conditions. Intelligent systems should be developed to enable such investigations. A good example was recently presented by Gheju and Balcu [33] investigating the efficiency of $Fe^0/MnO_2/H_2O$ for chromium removal. The idea behind this is that $MnO_2$ consumes $Fe^{II}$ from $Fe^0$ corrosion and delays material passivation (oxide scale in the vicinity of $Fe^0$) and particle cementation. In doing so, Gheju and Balcu [33] could perform batch experiments for up to 40 days. Another example is given by Noubactep et al. [120] using pyrite ($FeS_2$) to sustain $Fe^0$ reactivity in batch systems for up to 120 days. Characterizing the intrinsic reactivity of $Fe^0$ materials and using them in well-designed treatability studies is a good way to generate more reliable data which, in turn, will enable good modelling works.

### 3.4. Time for a Critical Discussion

The research presented herein deals with iron release from $Fe^0$. It can be argued that the presented information is not novel. However, the merit of these data is to be a kind of "general preliminary study" showing how differential iron is released from $Fe^0$ under various experimental conditions. Only when the scientific community is convinced on the limits of the current research approach, it will be ready to adopt alternatives for progress in knowledge. The aim is, thus, to enable or initiate a critical discussion.

The long scientific history of using "$Fe^0$ for water treatment" demonstrates that the evidence that iron corrosion generates adsorbing species for decontamination was known before 1850 [2] and has been independently rediscovered several times thereafter [4,6,119,121–123]. However, only in the context of subsurface reactive walls, $Fe^0$ was presented as its own reducing agent under environmental

conditions, a possible mistake that is still difficult to address [55–57,109,111]. It is very interesting in this regard to note that, parallel to the discovery that $Fe^0$ may be a reducing agent [124], Khudenko [70,123] has presented cementation as a tool to induce the reductive transformation of organic contaminants. In other words, the same material ($Fe^0$) was parallelly and independently presented as (i) its own reducing agent (electrons from $Fe^0$—direct reduction) [122,124] and a generator of reducing agents (electrons from $H_2$—indirect reduction) [70,123]. Moreover, the copresence of iron minerals and $Fe^{II}$ species in the $Fe^0/H_2O$ system implies the generation of structural $Fe^{II}$ that has been demonstrated to be a more powerful reducing agent than $Fe^0$ [125]. Clearly, even without considering the abundance of water (solvent), direct reduction appears unlikely to be quantitative. The following text from Khudenko [70] shall be used to illustrate the extent of the difficulty (references are modified/removed to facilitate readability):

"Reduction of organics by metals was described in 1913 by Clemmensen *(ref)*. Many such reactions are described by March *(ref, 1968)*. Sweeny (U.S. Patent No. 3,640,821) teaches to use zinc in acidic media for partial decomposition of dichlorodiphenyltrichloroethane DDT. Gillham (U.S. Patent No. 5,266,213) teaches to use metals placed underground for cleaning halogenated contaminants from groundwater. The Gillham process is also limited by anaerobic conditions in the treated water (Eh = $-100$ to $-200$ mV, absence of oxygen and other oxidants). Shifrin et al. [126] demonstrated nonaerated and aerated filters with a scrap steel for destruction of dyes. Contrary to Gillham, organics (COD) and color removals in this process improved with aeration... All elemental metal processes require the finely divided metals, preferably, metal dusts or sub millimeter size particles. Metal reduction of organics is driven by the potential difference between grains in the metal lattice. The magnitude of such difference is small (about 0.1 V). Accordingly, the organics transformation is very slow (hours to days) and incomplete (low removal of original compounds and high content of toxic residual products).

Another reading of this text from Khudenko [70] shows that the presence of $O_2$ and other oxidants, low pH values, the size of the metal particles and the presence of alloying elements impact the kinetics of contaminant transformations. The results presented herein depict considerable differences in the initial kinetics of Fe release from conventional materials which could be collectively called metallic iron by different investigators. The question arises which materials are used by individual researchers? Until this question is answered, controversy will remain the rule in the scientific literature. In essence, there is no fundamental difference between $Fe^0$-based bimetallic systems (e.g., Fe/Pd, Fe/Ni) and conventional $Fe^0$ materials for water treatment. Bimetallic systems solely enhance $Fe^0$ corrosion and, thus, accelerate the hydrogenation properties of the system [127].

Another important output of this work is that despite a century old history of technical applications [2,3,18,19], the $Fe^0$ technology is yet to be developed to the stage where it can be objectively compared to other technologies and/or where a realistic life cycle analysis (LCA) can be performed [128–130]. As the term suggests, such realistic LCAs will enable the identification of knowledge spillovers of $Fe^0$-based systems among technologies for water treatment [131–135].

## 3.5. A Framework for Future Research on the $Fe^0/H_2O$ System

Like electrocoagulation, the remediation $Fe^0/H_2O$ system can be considered an enigmatic technology [136]. Despite having been industrially used for over 140 years, there appears to be no science-based rules on the appropriate approaches for the design of individual applications [33,55, 102,103,111,137]. Accordingly, there is nothing like a path to the a priori modelling approach [86,87]. The root cause of this situation is certainly the fact that remediation using $Fe^0$ is a technology that lies at the intersection of four more fundamental technologies: (i) electrochemistry (generating adsorbents and flocs), (ii) adsorption, (iii) coagulation and (iv) flotation. Electrochemistry induces $Fe^0$ corrosion which implies water decontamination through adsorption, coprecipitation and size-exclusion. Adsorption is well-studied as a stand-alone technology. Coagulation and flotation inducing coprecipitation and size-exclusion are less well understood [138,139] but are also well-established. However, it is clear from the published literature that there is a lack of a quantitative appreciation of the way in which these

individual processes interact to provide efficient $Fe^0/H_2O$ systems [9,26,28,33,47,54–60,102,114,140] For the $Fe^0$-based systems to become an accepted and dependable water treatment technology and play the expected wider role in decentralized water treatments, research is required that focuses neither on simply making a specific pollutant-centred application work (treatability studies) nor on any one of the named foundation processes but rather on the emphasis of explaining and quantifying the key interactions between electrochemistry, adsorption, coprecipitation and size-exclusion. The $Fe^0/MnO_2/H_2O$ system is a good candidate to start this systematic work.

The results presented herein suggest that the $Fe^0/H_2O$ system's complexity can be simplified using a reductionist approach. Characterizing the initial iron dissolution in a diluted EDTA solution has shown diversities in reactivity for the eight tested $Fe^0$ materials. The tested pretreatment procedures also resulted in various responses for individual materials. Leaching iron from iron nails (FeN), iron filings (GI) and steel wool (SW) for 100 days has also revealed variabilities in reactivity in the long term. The question is how to continue this work to better understand the $Fe^0/H_2O$ system under field conditions? The example of the $k_{SA}$ concept [113] suggests that shortcut methods (correlation of experimental data via normalized parameters) will not help to design efficient and sustainable $Fe^0/H_2O$ systems. The complexity of the system and the myriad of interactions occurring therein suggest that it would be a mistake to pursue such a goal [60,136]. Future investigations should focus on quantifying the interactions that occur between the named relevant processes, starting with a range of model systems [107].

## 4. Conclusions

The current study used 30 $Fe^0/EDTA/H_2O$ systems to demonstrate the variability of the experimental designs used to investigate the $Fe^0/EDTA/H_2O$ systems. Each individual system is regarded as independent starting experimental conditions of a research which results are to be compared to other published data. It appears that a difference in the extent of Fe release of up to one order of magnitude (factor 8) was achieved while using quescient batch experiments (24 systems). The column experiments (6 systems) also demonstrated a significant variability in the reactivity of iron nails. The main feature of this research is that published data are not comparable and can be collectively regarded as qualitative. Accordingly, despite 30 years of intensive research, a systematic evaluation of the existing data is not possible. This means that relevant data leading to a site-specific design and operation of $Fe^0$ applications are still lacking. It is recommended that future research use $Fe^0/MnO_2/H_2O$-like systems to quantify the interactions between adsorption, coprecipitation and size-exclusion for the design of more reliable and sustainable systems. It is hoped that this approach will soon enable $Fe^0$ materials to play their expected role in establishing viable decentralized water treatment systems.

**Author Contributions:** R.H., A.I.N.-T., M.L., M.X., A.N.N., C.N. and K.N. contributed equally to the manuscript compilation and revisions.

**Funding:** This work is supported by the Ministry of Education of the People's Republic of China through the Program "Research on Mechanism of Groundwater Exploitation and Seawater Intrusion in Coastal Areas" (Project Code 20165037412) and by "the Fundamental Research Funds for the Central Universities" ("Research on the hydraulic tomographical method for aquifer characterization", Project Code 2015B29314). It is also supported by the Jiangsu Provincial Department of Education, Project Code 2016B1203503.

**Acknowledgments:** The manuscript was improved thanks to the insightful comments of anonymous reviewers from Sustainability. We acknowledge the support by the German Research Foundation and the Open Access Publication Funds of the Göttingen University.

**Conflicts of Interest:** The authors declare no conflict of interest.

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
