# Peer review of "The Impact of Selected Pretreatment Procedures on Iron Dissolution from Metallic Iron Specimens Used in Water Treatment"

_sustainability, doi:10.3390/su11030671_

Round 1

Reviewer 1 Report

This manuscript provides a much needed addition to the body of research, which is important in the field of metallic iron water treatment technologies. Therefore, I believe the paper is appropriate for publication in Sustainability, after addressing the following suggestions:

1) Line 2: Title:  ”The impact OF selected pretreatment ……”

2) Line 22: ” and pre-treatment procedure (socking in acetone, EDTA, H2O and HCl for 17 hours)”

Please add inside the brackets also the NaCl pre-treatment!

3) Line 122: Table 1:

- Why is the mass (the ”m” value) of GI2 not available? It should be known because the GI2 sample was weighted by you with a balance, just like the other GI samples.

- Why is the ”P” value of GI2 and GI3 not available?

- You mentioned that ”The N value was determined for spherical GI3 because filings are difficult to count”. However, I see that for GI1 N ˃ 232, while for GI2 N = n.a. I believe that, because the number of GI1 and GI2 grains making up the 2.0 g of Fe0 is difficult to count, the N value should be the same for both: either ˃ 232, or n.a.

4) Lines 131-132: ”The pre-treated materials was further washed three times with tap water prior to the addition of the EDTA solution”

I believe it would be very interesting to know if and how was the pre-treated material dried after the washing with tap water. Or, the EDTA solution was added over the wet pre-treated material? Please give some more details.

5) Line 250: Table 2: Please explain what is the difference between the two Fe(II) concentrations (column 3 and 4)

6) Line 298 and 300: please replace ”Table 3” with ”Table 2”.

7) Line 356: ” Fe0 is oxidized by water and molecular O2 by FeII”.

Please replace : ”molecular O2 by FeII” with ”FeII by molecular O2”.

Author Response

This manuscript provides a much needed addition to the body of research, which is important in the field of metallic iron water treatment technologies. Therefore, I believe the paper is appropriate for publication in Sustainability, after addressing the following suggestions:

Many thanks for this evalation!

1) Line 2: Title: ”The impact OF selected pretreatment ……” Done, thanks!

2) Line 22: ” and pre-treatment procedure (socking in acetone, EDTA, H2O and HCl for 17 hours)”

Please add inside the brackets also the NaCl pre-treatment! Done, thanks!

3) Line 122: Table 1:

- Why is the mass (the ”m” value) of GI2 not available? It should be known because the GI2 sample was weighted by you with a balance, just like the other GI samples.

GI2 is not used in column experiements, only GI1 was used. The mass of GI3 is given to evaluate the N value for GI1. The title of the table is revised! Please consider that the table summarized the characteristics (all specimens) and some experimentals results (materials tested in columns).

- Why is the ”P” value of GI2 and GI3 not available?

These two specimens were not used in column experiments.

- You mentioned that ”The N value was determined for spherical GI3 because filings are difficult to count”. However, I see that for GI1 N ˃ 232, while for GI2 N = n.a. I believe that, because the number of GI1 and GI2 grains making up the 2.0 g of Fe0 is difficult to count, the N value should be the same for both: either ˃ 232, or n.a.

The reviewer is right, but the evaluation of N for GI2 was not necessary as it was not used in column studies.We have changed 'n.a.' by '> 232' for GI2 because it is certain, Then we have written 2.0 because it is simple logic. Thanks for this important remarks!

4) Lines 131-132: ”The pre-treated materials was further washed three times with tap water prior to the addition of the EDTA solution”

I believe it would be very interesting to know if and how was the pre-treated material dried after the washing with tap water. Or, the EDTA solution was added over the wet pre-treated material? Please give some more details.

Many thanks, the following sentence is added: 'The EDTA solution was added to the wet pre-treated materials'

5) Line 250: Table 2: Please explain what is the difference between the two Fe(II) concentrations (column 3 and 4)

Thanks, the second value represents the standard deviation of individual triplicates. This is know specified in the legend.

6) Line 298 and 300: please replace ”Table 3” with ”Table 2”. Done, thanks!

7) Line 356: ” Fe0 is oxidized by water and molecular O2 by FeII”.

Please replace : ”molecular O2 by FeII” with ”FeII by molecular O2”. Done, many thanks!

Reviewer 2 Report

The research topic of the paper is interesting but the manuscript needs minor revisions before its potential publication.

First, the objective of the analysis should be discussed in a more detailed way in the abstract and introduction sections.

Second, the literature review should be enriched in such a way that the knowledge spillovers for water treatment effects are well identified (Aldieri & Vinci, 2017; Hayek & Stejskal, 2018).

Finally, the results should be discussed and improved in terms of policy implications.

References.

Aldieri, L. & Vinci, C. P. (2017). The Role of Technology Spillovers in the Process of Water Pollution Abatement for Large International Firms. Sustainability, 9(5), 868.

Hayek, P. & Stejskal, J. (2018). R&D Cooperation and Knowledge Spillover Effects for Sustainable Business Innovation in the Chemical Industry. Sustainability, 10(4), 1064.

Author Response

The research topic of the paper is interesting but the manuscript needs minor revisions before its potential publication.

Many thanks for this evaluation!

First, the objective of the analysis should be discussed in a more detailed way in the abstract and introduction sections.

We are rather surprised by this pertinent remarks! Indeed the context of the research is almost completely missed (in the Abstract to 100 %) in both parts.

The abstract now starts with: 'Studies were undertaken to determine the reasons why published information regarding the efficiency of metallic iron (Fe0) for water treatment is conflicting and even confusing.'

We are convinced that this addition alone makes the whole more comprehensive. However, the Introduction is also modified and the addition is blue-marked.

Second, the literature review should be enriched in such a way that the knowledge spillovers for water treatment effects are well identified (Aldieri & Vinci, 2017; Hayek & Stejskal, 2018).

We have added the following section in the discussion:

Another important output of this work, is that despite a century old history of technical application [2,3,18,19], the Fe0 technology is yet to be developed to the stage where it can be objectively compared to other technologies and/or a realistic life cycle analysis (LCA) can be performed [128-130]. At the term, such realistic LCAs will enable the identification of knowledge spillovers of Fe0-based systems among technologies for water treatment [131-135].

Finally, the results should be discussed and improved in terms of policy implications.

We agree with the reviewer, but the message is fmainly for the research community. We have checked the possibility of questioning the funding policy but it is not universal as for example what we missed in Germany is available in China or elsewhere.